# High-potency ligands for DREADD imaging and activation in rodents and monkeys

Jordi Bonaventura [1,13], Mark A.G. Eldridge [2,13], Feng Hu [3,13], Juan L. Gomez[1], Marta Sanchez-Soto[4], Ara M. Abramyan[5], Sherry Lam[1], Matthew A. Boehm[1], Christina Ruiz[6], Mitchell R. Farrell[6], Andrea Moreno[7], Islam Mustafa Galal Faress[7], Niels Andersen[7], John Y. Lin [8], Ruin Moaddel[9], Patrick J. Morris[10], Lei Shi [5], David R. Sibley [4], Stephen V. Mahler [6], Sadegh Nabavi[7], Martin G. Pomper[3], Antonello Bonci[11], Andrew G. Horti[3], Barry J. Richmond [2] & Michael Michaelides [1,12]

Designer Receptors Exclusively Activated by Designer Drugs (DREADDs) are a popular chemogenetic technology for manipulation of neuronal activity in uninstrumented awake animals with potential for human applications as well. The prototypical DREADD agonist clozapine N-oxide (CNO) lacks brain entry and converts to clozapine, making it difficult to apply in basic and translational applications. Here we report the development of two novel DREADD agonists, JHU37152 and JHU37160, and the first dedicated [18]F positron emission tomography (PET) DREADD radiotracer, [[18]F]JHU37107. We show that JHU37152 and JHU37160 exhibit high in vivo DREADD potency. [[18]F]JHU37107 combined with PET allows for DREADD detection in locally-targeted neurons, and at their long-range projections, enabling noninvasive and longitudinal neuronal projection mapping.

[1] Biobehavioral Imaging and Molecular Neuropsychopharmacology Unit, National Institute on Drug Abuse Intramural Research Program, Baltimore, MD 21224, USA. [2] Laboratory of Neuropsychology, National Institute of Mental Health Intramural Research Program, Bethesda, MD 20892, USA. [3] Department of Radiology Johns Hopkins School of Medicine, Baltimore, MD 21205, USA. [4] Molecular Neuropharmacology Section, National Institute of Neurological Disorders and Stroke Intramural Research Program, Bethesda, MD 20814, USA. [5] Computational Chemistry and Molecular Biophysics Unit, National Institute on Drug Abuse Intramural Research Program, Baltimore, MD 21224, USA. [6] Department of Neurobiology & Behavior, University of California, Irvine, CA 92697, USA. [7] Department of Molecular Biology and Genetics, Dandrite, Aarhus University, 8000 Aarhus C, Aarhus, Denmark. [8] School of Medicine, College of Health and Medicine, University of Tasmania, Tasmania, TAS 7000, Australia. [9] Laboratory of Clinical Investigation, National Institute on Aging Intramural Research Program, Baltimore, MD 21224, USA. [10] National Center for Advancing Translational Sciences, Rockville, MD 20850, USA. [11] Synaptic Plasticity Section, National Institute on Drug Abuse Intramural Research Program, Baltimore, MD 21224, USA. [12] Department of Psychiatry, Johns Hopkins School of Medicine, Baltimore, MD 21205, USA. [13] These authors contributed equally: Jordi Bonaventura, Mark A. G. Eldridge, Feng Hu. Correspondence and requests for materials should be addressed to A.G.H. (email: ahorti1@jhmi.edu) or to B.J.R. (email: barryrichmond@mail.nih.gov) or to M.M. (email: mike.michaelides@nih.gov)

Designer receptors exclusively activated by designer drugs (DREADD)[1] technology is a powerful chemogenetic approach used for neuromodulation in uninstrumented research animals. Combining a DREADD with translational molecular imaging methods such as positron emission tomography (PET) could extend this technique to clinical applications, by providing a means for noninvasive confirmation of receptor expression and function. DREADD ligands developed to date have characteristics that limit their utility for translational central nervous system (CNS) applications[2]. The prototypical DREADD agonist, clozapine N-oxide (CNO), has poor brain penetrance and, via metabolic degradation, gives rise to the antipsychotic drug clozapine, which is the main active in vivo CNS DREADD agonist[2]. Therefore new, potent DREADD agonists and selective, high-affinity DREADD PET radioligands are needed to advance the translational potential of this powerful chemogenetic technology. Here, we use an array of complementary in vitro, ex vivo and in vivo approaches in rodents and in monkeys to report the development of JHU37152 and JHU37160, the first DREADD agonists with high in vivo potency for CNS applications. We also report the development of the first [18F]-labeled high-affinity DREADD PET radioligand, [18F]JHU37107, which enables noninvasive and longitudinal DREADD detection and localization in locally targeted neurons and at long-range projection sites. Together, these new tools expand the power of DREADD chemogenetic technology to encompass translational applications for noninvasive manipulation and visualization of neuronal circuits.

## Results

### New DREADD ligands with high in vitro affinity and potency.
Recently, a new second-generation DREADD ligand, Compound 21 (C21), was put forward as an effective DREADD agonist with excellent brain penetrance that does not convert to clozapine[3–5]. However, there have not yet been reports of the use of C21 in nonhuman primates (NHP). To determine the extent to which C21 is suitable for activating DREADDs, we carried out a comprehensive series of studies in rodents and rhesus monkeys described in Supplementary Figs. 1 and 2. Collectively, our findings suggest that C21, like CNO, exhibits lower in vivo DREADD potency than clozapine and is particularly not efficient in NHP applications.

To search for potent DREADD agonists suitable for in vivo biological studies, we would like to identify new ligands with good performance. To this end, we profiled two other compounds from the series described by Chen et al.[3]. Compound **13** (C13) and Compound **22** (C22) (Fig. 1a). Both C13 and C22 exhibited high-DREADD affinity and C13 had twofold higher affinity ($^{hM3Dq}K_i = 4.3$ nM; $^{hM4Di}K_i = 4.3$ nM) than C22 ($^{hM3Dq}K_i = 11$ nM; $^{hM4Di}K_i = 9$ nM) (Fig. 1b). Correspondingly with its binding affinity, C13 (10 nM) was able to displace [3H]clozapine selectively from hM3Dq but not from endogenous clozapine-binding sites (Fig. 1c, d). Next, we synthesized [3H]C13, which bound DREADDs expressed in brain tissue sections at very low concentrations, exhibited greater DREADD selectivity than [3H]clozapine (particularly in the macaque) (Fig. 1e–h) and a promising peripheral biodistribution profile in mice (Supplementary Fig. 3). [3H]C13 exhibited high ex vivo DREADD engagement in dopamine D1 receptor (D1)-DREADD mice (Fig. 1i) indicating high brain bioavailability and direct DREADD engagement.

The PET radioligand [11C]clozapine has been used to image DREADDs in vivo in rodents and in NHPs[2,6,7]. However, the short half-life (~20 min) of the [11C] radionuclide does not permit combined use of chemogenetics with PET at institutions that lack cyclotrons and thus limits its overall research and potential clinical

use. The [18F] radionuclide, with a half-life of ~110 min, allows for commercialization, extends the use of PET to chemogenetics applications at institutions that lack cyclotrons, and facilitates imaging at longer time intervals. To make an [18F]-labeled PET DREADD ligand, we designed and synthesized fluorinated analogs of C13 and C22. We reasoned that the presence of the additional fluorine would make it simple to radiolabel the compound with [18F] via direct substitution, and by exploring various potential sites for fluorination we would be able to preserve, or perhaps even improve DREADD affinity. We identified three analogs, JHU37107 ($^{hM3Dq}K_i = 10.5$ nM; $^{hM4Di}K_i = 23.5$ nM), JHU37152 ($^{hM3Dq}K_i = 1.8$ nM; $^{hM4Di}K_i = 8.7$ nM), and JHU37160 ($^{hM3Dq}K_i = 1.9$ nM; $^{hM4Di}K_i = 3.6$ nM) displaying the highest in vitro DREADD affinity (Fig. 1j–l). We then docked the highest-affinity JHU37160 in the ligand binding pocket of an hM4Di model and identified a stable pose of JHU37160 using molecular dynamics simulations (Fig. 1m). In the selected pose, the pyramidal nitrogen of JHU37160 forms an ionic interaction with Asp112[3.32], and its ethyl group makes favorable hydrophobic-aromatic interactions with Tyr416, Tyr439, and Tyr443, which likely contribute to the ~25-fold improved affinity of JHU37160 compared to C21. The chloride group of JHU37160 forms a halogen bond with the backbone carbonyl oxygen of Gly203 (note Gly203 is one of the two mutations in DREADD). Based on this pose, we predicted that fluorination at the para positions would be well tolerated, which was indeed the case for JHU37152 and JHU37160.

We tested both compounds for in situ [3H]clozapine displacement in brain tissue from WT and D1-DREADD mice and found that both JHU37152 and JHU37160 exhibited selective [3H] clozapine displacement from DREADDs and not from other clozapine-binding sites at concentrations up to 10 nM (Fig. 1n, o). In addition, both compounds were potent DREADD agonists with high potency and efficacy in fluorescent and BRET-based assays in HEK-293 cells; JHU37152: ($^{hM3Dq}EC_{50}$: 5 nM; $^{hM4Di}EC_{50}$: 0.5 nM) and JHU37160: ($^{hM3Dq}EC_{50}$: 18.5 nM; $^{hM4Di}EC_{50}$: 0.2 nM) (Fig. 1p, q), whereas no responses were observed in untransfected HEK-293 cells (Supplementary Fig. 4).

### JHU37152 and JHU37160 exhibit high in vivo DREADD occupancy.
In contrast to CNO and C21, mice injected (IP) with a 0.1 mg kg−1 dose of either JHU37152 or JHU37160 (Fig. 2a) showed high brain to serum concentration ratios (~eightfold higher in the brain than serum at 30 min), indicating active sequestration in brain tissue (Fig. 2b). Neither JHU37152 nor JHU37160 were P-gp substrates (Supplementary Fig. 5). The CSF concentration of JHU37160 at this same dose in the monkey was below our system's detection limit. However, JHU37160 was detected in serum where it showed a similar profile as in the mouse (Supplementary Fig. 6). At this same dose, 0.1 mg kg−1, JHU37152 and JHU37160 occupied approximately 15–20% of striatal DREADDs in mice (Fig. 2c, d). In rats, 0.1 mg kg−1 JHU37160 occupied approximately 80% of cortical hM4Di (Fig. 2e, f). In monkey, 0.1 mg kg−1 JHU37160 (and to a lesser extent, 0.01 mg kg−1) produced [11C]clozapine DREADD displacement at hM4Di expressed in the amygdala (Fig. 2g, h, and Supplementary Fig. 6).

### JHU37152 and JHU37160 exhibit high in vivo DREADD potency.
As predicted from the above findings, JHU37152 and JHU37160 were potent in vivo DREADD agonists, selectively inhibiting locomotor activity in D1-hM3Dq and D1-hM4Di mice at doses ranging from 0.01 to 1 mg kg−1 without any significant locomotor effects observed at these doses in WT mice (Fig. 3a–c).

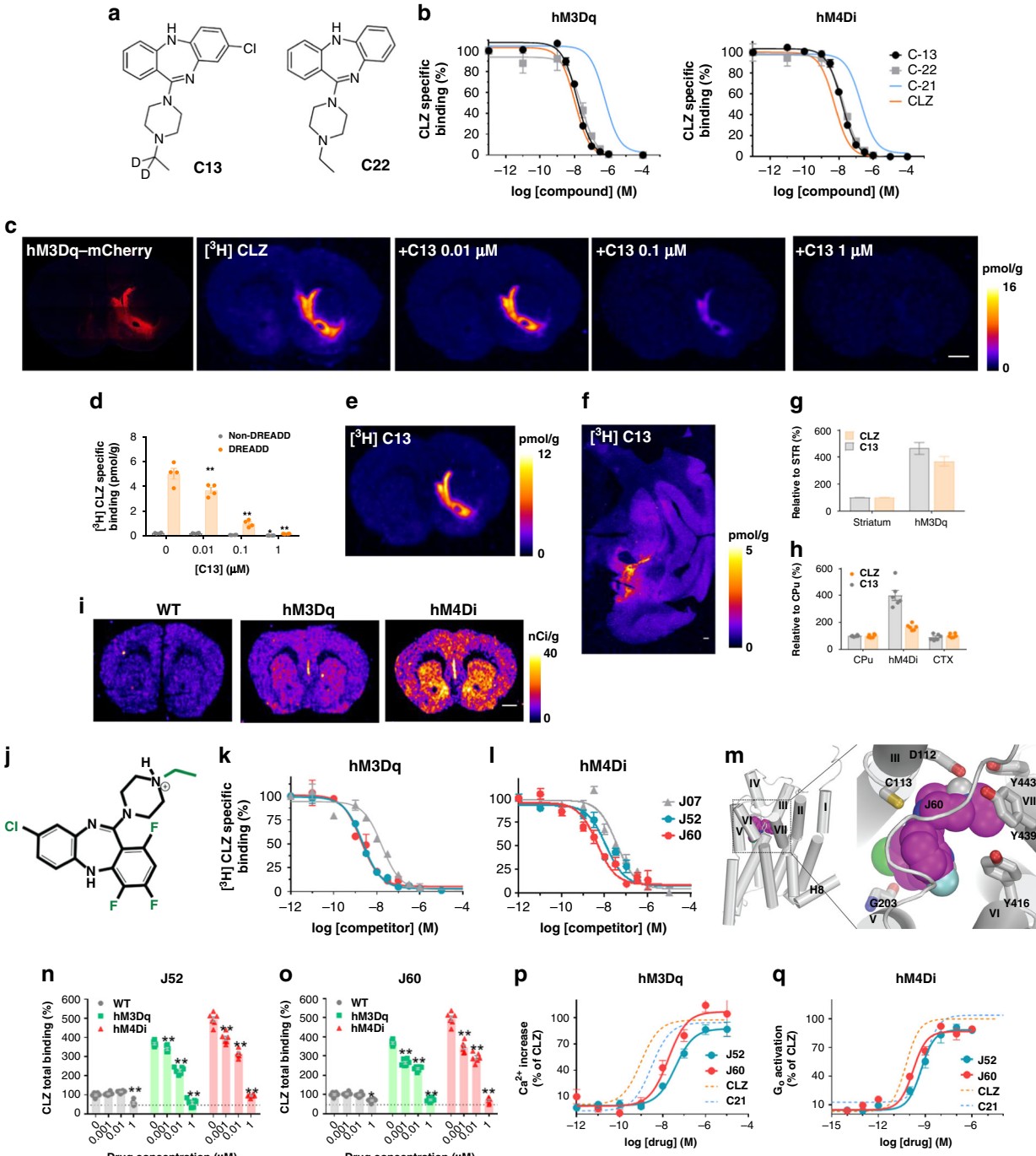

**Fig. 1** New DREADD ligands displaying high in vitro DREADD affinity and potency. **a** Compound 13 (C13) and Compound 22 (C22) structures. **b** Binding competition curves of [³H]CLZ versus increasing concentrations of C13 and C22 in HEK-293 cells expressing DREADDs. C13 and C22 exhibit comparable DREADD affinity to clozapine (CLZ) with C13 showing ~twofold greater affinity than C22. CLZ and C21 competition curves from Supplementary Fig. 1 are overlaid for comparison. **c, d** C13 selectively blocks [³H]CLZ binding to DREADDs in mouse slices at 10 nM. Representative images of sections collected from 3 different mice are displayed and quantified in (**d**) as mean ± SEM. Two-way ANOVA followed by Dunnett's test, $^*p < 0.05$ and $^{**}p < 0.01$ compared with the respective vehicle. **e–h** [³H]C13 binds with greater selectivity than [³H]CLZ to DREADDs in mouse and monkey brain tissue expressing AAV-hM3Dq and AAV-hM4Di, respectively. **i** Intraperitoneal (IP) injection of [³H]C13 readily enters the brain and accumulates in DREADDs expression areas in D1-DREADD mice. Representative images from 3 mice per condition. **j–l** JHU37107 (J07), JHU37152 (J52), and JHU37160 (J60) are high-affinity DREADD ligands. **m** Docking and molecular dynamics simulation of J60 in the ligand binding pocket of a hM4Di model. **n, o** J60 and J52 selectively displace [³H]CLZ at a concentration of 1 and 10 nM from hM3Dq and hM4Di expressed in mouse brain sections ($n = 3$ mice per condition). **p, q** J60 and J52 activate hM3Dq and hM4Di expressed in HEK293 cells with high potency (experiments performed 3–5 times). In all cases, data are represented as mean ± SEM. Scale bars are 1 mm. Source data are provided as a Source Data file

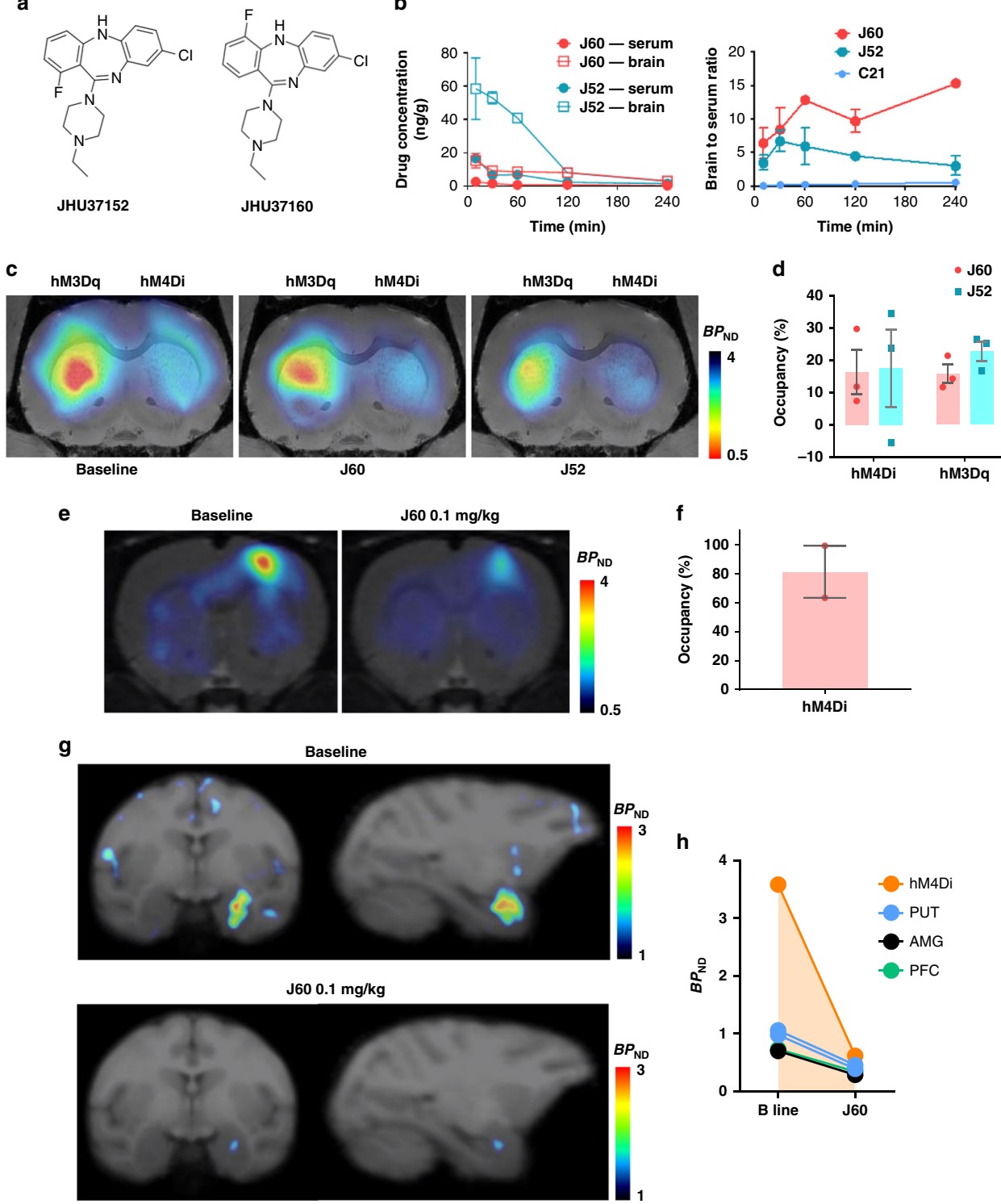

**Fig. 2** JHU37152 and JHU37160 exhibit high in vivo DREADD occupancy. **a** Structures of JHU37152 (J52) and JHU37160 (J60). **b** Brain and serum concentrations and ratios of J52 and J60 in mice ($n = 4$ mice per condition) at different time points after a 0.1 mg kg$^{-1}$ (IP) injection. C21 (1 mg kg$^{-1}$, IP) data are same as shown in Supplementary Figures for comparison purposes. **c**, **d** J52 and J60 (0.1 mg kg$^{-1}$, IP) displace in vivo [$^{11}$C]clozapine binding to DREADDs in AAV-DREADD-expressing mice ($n = 5$ mice). **e**, **f** J60 (0.1 mg kg$^{-1}$, IP) selectively blocks in vivo [$^{11}$C]clozapine binding to DREADDs in rats ($n = 3$ rats). **g**, **h** J60 (0.1 mg kg$^{-1}$) blocks in vivo [$^{11}$C]clozapine binding to hM4Di in the monkey. All data represented are mean ± SEM except in (**h**) where individual values are displayed. Source data are provided as a Source Data file

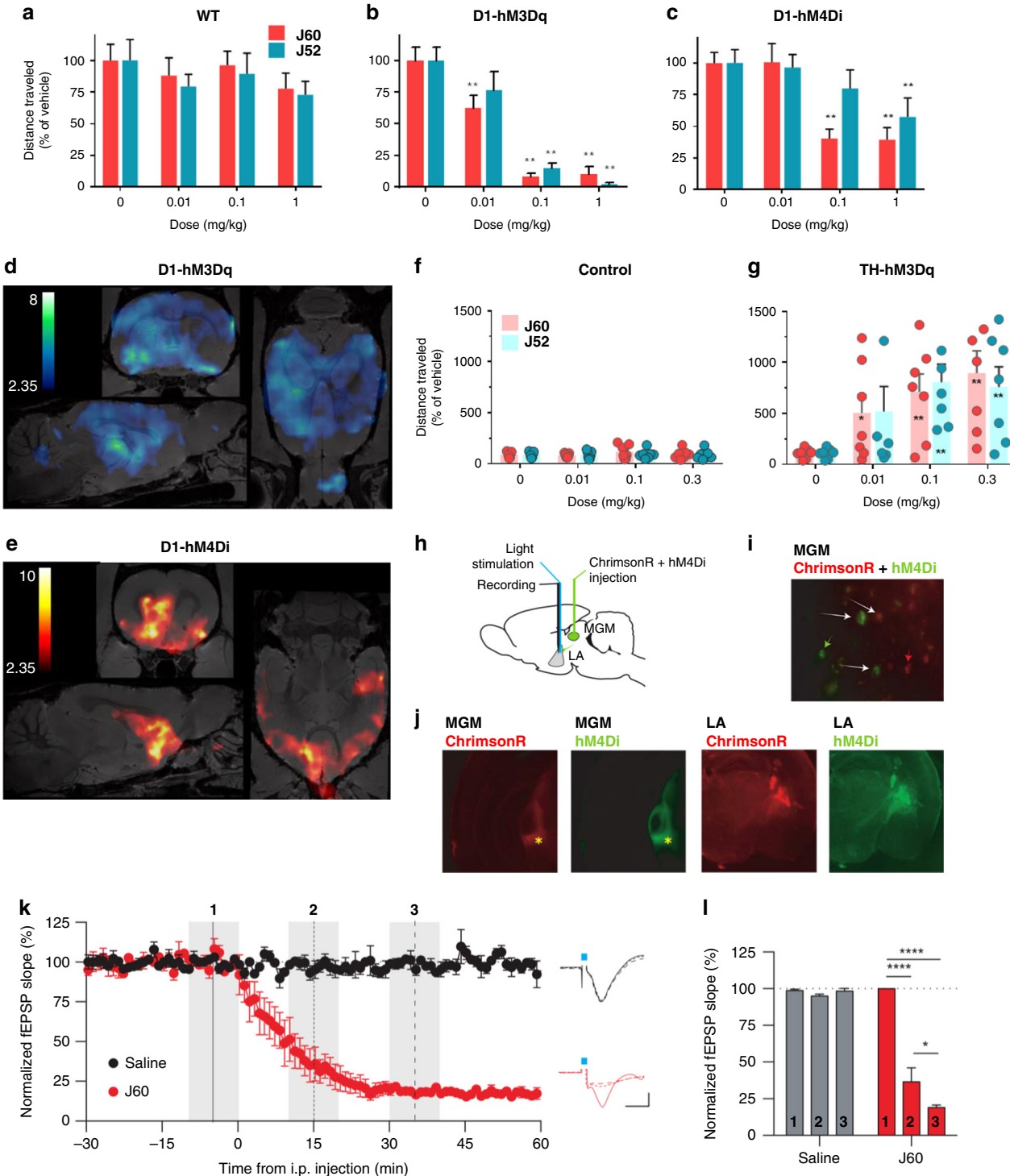

**Fig. 3** JHU37152 and JHU37160 exhibit high in vivo DREADD potency. **a–c** J60 and J52 produce potent inhibition of locomotor activity in transgenic D1-DREADD mice but not in wild-type (WT) mice ($n = 7$ to 19 mice per condition). Two-way repeated measures ANOVA followed by Dunnett's multiple comparison tests were performed, $^*p < 0.05$ and $^{**}p < 0.01$ compared with the respective vehicle. **d, e** DREADD-assisted metabolic mapping (DREAMM) using [$^{18}$F]FDG in D1-hM3Dq and D1-hM4Di mice ($n = 4$ mice per condition) reveals opposing and differential recruitment of whole-brain functional networks. **f, g** J52 and J60 produce potent activation of locomotor activity in rats ($n = 7$ rats per condition) expressing hM3Dq in tyrosine hydroxylase (TH)-expressing neurons in the ventral tegmental area. One-way repeated measures ANOVA followed by Dunnett's multiple comparison tests were performed, $^*p < 0.05$ and $^{**}p < 0.01$ compared with the respective vehicle. **h–j** Design of in vivo electrophysiological experiment and IHC showing hM4Di (green) and ChrimsonR (red) expression in the medial division of the medial geniculate nucleus (MGM) and lateral amygdala (LA). **k, l** J60 (0.1 mg kg$^{-1}$) produces rapid and potent hM4Di-driven inhibition of light-evoked neuronal activation. Data are represented as mean ± SEM, $^*p < 0.05$, $^{***}p < 0.001$. Source data are provided as a Source Data file

Using DREADD-assisted metabolic mapping (DREAMM)[8,9] with the [18F]fluorodeoxyglucose (FDG) tracer to assess changes in regional brain activity after hM3Dq or hM4Di activation of D1 neurons, 0.1 mg kg$^{-1}$ (IP) JHU37160 produced metabolic changes in distinct and largely nonoverlapping brain networks in D1-hM3Dq (Fig. 3d) versus D1-hM4Di (Fig. 3e) mice and caused no significant brain metabolic changes in WT mice (Supplementary Fig. 7). The recruitment of distinct, almost mutually exclusive networks was paralleled by metabolic changes with opposite directionality upon differential modulation of D1 neurons with hM3Dq and hM4Di: decreased metabolism in D1-hM3Dq and increased metabolism in D1-hM4Di mice, effects likely mediated via activation and inhibition of striatal GABAergic D1-expressing neurons respectively.

In a competitive binding screen, JHU37152 and JHU37160 exhibited lower affinity than clozapine at 5-HT receptors (Supplementary Fig. 8). Although the overall target profile of both compounds was similar to clozapine, they did not produce any agonist effect in functional assays performed in HEK-293 cells lacking DREADDs, but expressing endogenous clozapine-binding targets[10] (Supplementary Fig. 4). As such, they are expected to behave as antagonists at these receptors, competing with endogenous neurotransmitters at these same binding sites. In contrast, JHU37152 and JHU37160 evidenced DREADD activation at lower concentrations than clozapine, indicating that the former compounds are more selective DREADD agonists.

In TH-hM3Dq rats, 0.01–0.3 mg kg$^{-1}$ JHU37152 and JHU37160 led to robust, selective increases in hM3Dq-stimulated locomotion (Fig. 3f, g). To further characterize the performance of JHU37160 as an in vivo DREADD agonist, we performed in vivo electrophysiology experiments in which hM4Di was co-expressed with a new light-drivable channelrhodopsin, ChrimsonR (Supplementary Fig. 9), in the terminals of the medial division of the medial geniculate nucleus (MGM) to striatum/lateral amygdala (LA) pathway (Fig. 3h–j) in mice. Mice were implanted with optrodes in the LA. A dose of 0.1 mg kg$^{-1}$ (IP) JHU37160 elicited rapid inhibition of ChrimsonR-induced terminal activation; 60% inhibition was observed at ~10 min (36 ± 9% of baseline), and maximal inhibition at 30 min after injection (19 ± 2% of baseline) (Fig. 3k, l). These effects were hM4Di-dependent; the same injection of JHU37160 had no effect on electrical stimulation evoked responses in animals without DREADD expression (Supplementary Fig. 10).

**[18F]JHU37107 enables noninvasive neuronal projection mapping.** The high-affinity profiles of JHU37152, JHU37160, and JHU37107 stimulated efforts to develop them into [18F]-labeled PET imaging probes. Unfortunately, the discrete positions of the fluorine atoms in JHU37152 and JHU37160 made radiosynthesis efforts challenging and inefficient. The most radiochemically favorable structure was [18F]JHU37107 which we radiolabeled with high yield, molar activity and radiochemical purity (Fig. 4a). In D1-DREADD transgenic mice, [18F]JHU37107 exhibited robust uptake in DREADD-expressing brain regions as compared to areas devoid of DREADD expression (Fig. 4b–d). This signal was displaceable by 0.1 mg kg$^{-1}$ (IP) JHU37160 (Fig. 4b–d) indicating specific binding of [18F]JHU37107 at DREADD sites. We also tested [18F]JHU37107 in rats with unilateral hM3Dq (Fig. 4e) or hM4Di (Fig. 4f–j) expression in the right motor cortex. [18F]JHU37107 permitted hM4Di visualization in both the AAV injection site as well as at known proximal and distal anatomical projection sites such as striatum, contralateral cortex, and motor thalamus. The in vivo localization of the [18F]JHU37107 signal matched the ex vivo expression of DREADDs established by post hoc immunohistochemistry staining

(Fig. 4f–j). Finally, we tested [18F]JHU37107 in a monkey expressing hM4Di in the right amygdala (Fig. 4k). [18F]JHU37107 exhibited favorable pharmacokinetic properties (Fig. 4l, m) and metabolite profile (Supplementary Fig. 11) in this species. More importantly, it was able to directly label hM4Di receptors (Fig. 4k–m), allowing robust detection of DREADDs with a dedicated [18F]-labeled radioligand for the first time in nonhuman primates.

**Discussion**

The human muscarinic receptor-based DREADDs are the most popular chemogenetic technology for basic research and are used by a large number of laboratories around the world. Although the majority of DREADD use has been in rodents, DREADDs have also been applied for experimental use in monkeys recently[6,11–13], a critical step before human translation. Results from prior studies[2,5,14], and now from the current study, indicate that the DREADD agonists developed to date, while efficacious in certain applications, do not display sufficient potency or selectivity in others. In rodents (rats and mice), we show here that C21 activated hM3Dq at doses as low as 0.1 mg kg$^{-1}$, however, it was less potent at activating hM4Di, which required at least 1 mg kg$^{-1}$. In mice without DREADDs, doses higher than 1 mg kg$^{-1}$ produced off-target effects. Furthermore, in WT mice, the 1 mg kg$^{-1}$ dose of C21 required to activate hM4Di produced changes in brain metabolic activity (FDG uptake) even though we and others[5] did not detect any behavioral effects using this dose. In contrast, an equipotent dose of clozapine (0.1 mg kg$^{-1}$) did not produce any significant changes in brain metabolic activity. In monkeys, the minimal doses required to achieve DREADD occupancy also extensively displaced [11C]clozapine from endogenous targets and produced nonspecific effects[11]. In summary, C21 has a small window of selectivity to activate hM4Di in rodents, which may potentially be compensated by overexpression of the DREADD receptor, and furthermore displays a wide range of off-target effects at the minimal hM4Di-effective doses in monkeys.

In addition to the characterization of C21, here we report the development of a new set of DREADD agonists that exhibit high in vivo potency and CNS DREADD occupancy in both rodents and in old world monkeys. While their selectivity is not ideal (i.e., comparable to clozapine), their high in vivo potency allows for dose adjustments with minimal off-target effects and importantly they exhibit promising characteristics for DREADD use in monkeys. Our data suggest that further steps to improve selectivity require divergence from the dibenzodiazepine (clozapine-based) scaffold and/or require new rationally engineered mutations in the DREADD binding pocket to differentiate it from endogenous wild-type receptors.

The other notable advance in this study is the development of the first, high-affinity [18F]-labeled DREADD PET ligand. The use of [18F], with six times longer half-life than [11C], allows for this ligand to be shipped to facilities without cyclotrons or that lack the necessary radiosynthesis infrastructure and capabilities. Moreover, it offers the possibility to scan several animals using one synthesis or to perform longer scans for extensive kinetic modeling and occupancy studies. Finally, this new PET ligand provides strong somatic signaling of receptor expression in both rodents and monkeys, and in rodents, at least, there is signal that represents projections to remote locations from the primary viral injection site, making noninvasive and longitudinal visualization of cell type-specific neuroanatomical projections possible in the living mammalian subject.

Chemogenetic technologies, like DREADDs confer the ability to manipulate neuronal activity across distributed brain circuits without the need for implantable devices[15], thereby making them

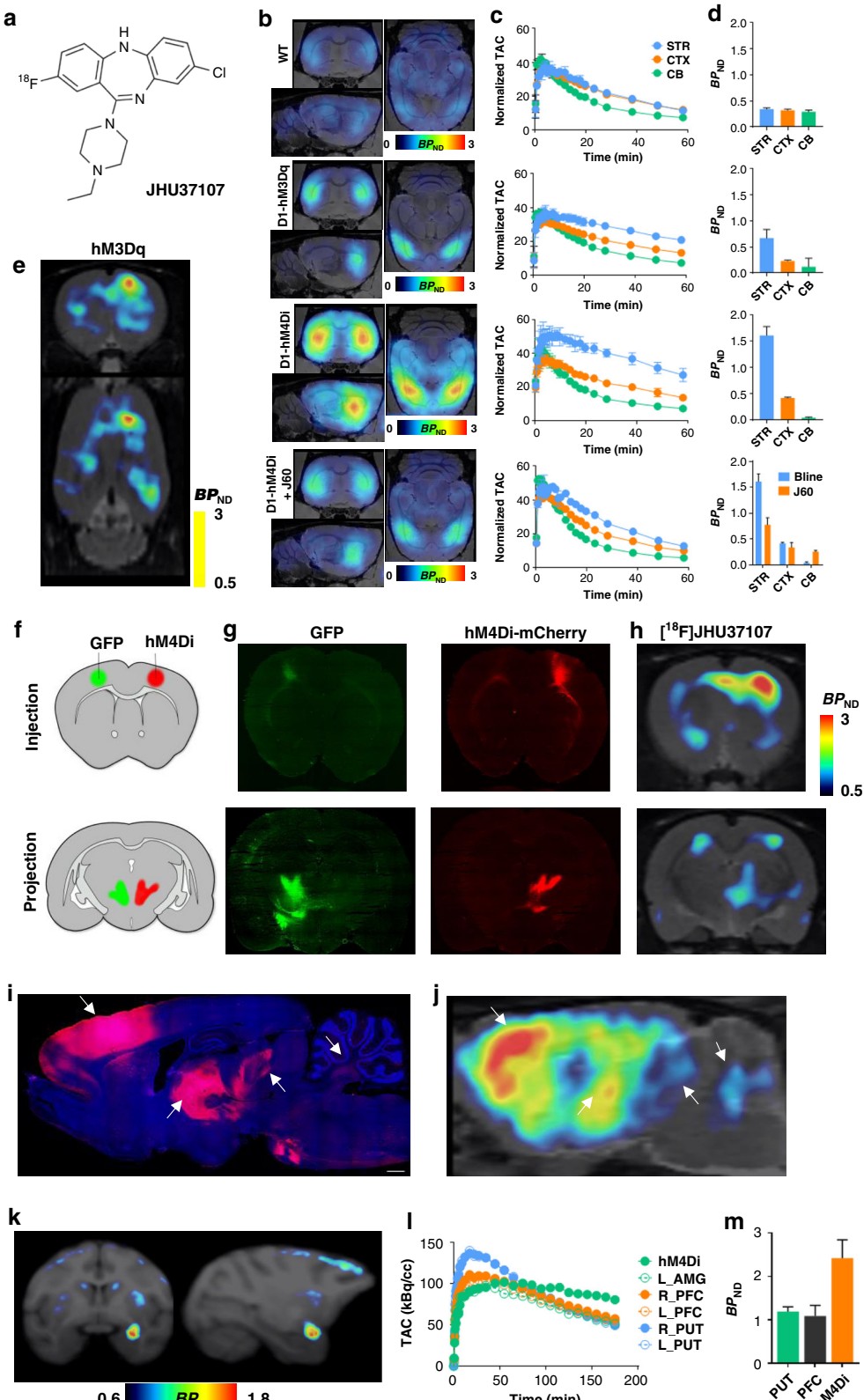

**Fig. 4** [18F]JHU37107 enables noninvasive detection of DREADD in locally-targeted cells and at their long-range projections. **a** Structure of [18F]JHU37107. **b–d** [18F]JHU37107 selectively binds to DREADDs in the brain of transgenic D1-DREADD mice ($n = 3$ mice per condition) and is blocked by 0.1 mg kg$^{-1}$ of JHU37160. **e–j** [18F]JHU37107 selectively binds to AAV-DREADDs expressed in the rat cortex and enables noninvasive and longitudinal mapping of both local (injection site) and long-range projections of motor cortex circuitry (ventrolateral thalamus shown as a main hub). Representative immunohistochemical images showing GFP (green) or HA-tagged DREADDs (red) from representative rats are shown side by side with their corresponding [18F]JHU37107 PET images. The white arrows point at corresponding anatomical regions. **k–m** [18F]JHU37107 binds to hM4Di expressed in the monkey amygdala and at putative projection sites. All data are represented as mean ± SEM except in (**l**) and (**m**) where individual values are displayed. Scale bars are 1 mm. Source data are provided as a Source Data file and the raw PET data are available upon request

especially useful in awake and even unrestrained animals. The development of [18F]JHU37107 and our new agonists provide means to perform diagnostic imaging of DREADDs in neurotherapeutic contexts,—i.e., "neurotheranostics"[16], making it possible to imagine future development of cell type- and circuit-specific neuromodulation for humans. In the same manner, since FDG-PET is a routine human procedure, DREAMM[8,9] can also be used to evaluate, longitudinal, noninvasive assessment of whole-brain, functional circuit activity as a function of chemogenetic-based therapies. In sum, if the novel pharmacological tools and approaches we describe here are extended to humans, DREADD-based neurotheranostics[16] would comprise a novel precision-medicine approach that could be used for developing chemogenetic-based cell type- and circuit-specific neuromodulation for the precision or personalized treatment of various brain disorders.

## Methods

**Experimental subjects**. Wild-type mice (C57BL/6J) were ordered from Jackson Laboratories and rats (Sprague–Dawley) were ordered from Charles River. Rodents were male and ordered at ~6 weeks of age. Transgenic mice were bred at NIDA breeding facility. Transgenic mice expressing the enzyme cre recombinase under the control of the dopamine D1 receptor promoter (D1-Cre, FK150 line, C57BL/6J congenic, Gensat, RRID: MMRRC_036916-UCD) were crossed with transgenic mice with cre recombinase-inducible expression of hM4Di DREADD (R26-hM4Di/mCitrine, Jackson Laboratory, stock no. 026219) or hM3Dq DREADD (R26-hM3Dq/mCitrine, Jackson Laboratory, stock no. 026220). Three male rhesus monkeys (*Macaca mulatta*) weighed 8–12 kg. All experiments and procedures complied with all relevant ethical regulations for animal testing and research and followed NIH guidelines and were approved by each institute's animal care and use committees.

**Cell culture and transfection**. Human embryonic kidney (HEK-293, ATCC) cells were grown in Dulbecco's modified Eagle's medium (DMEM; Gibco, Thermo-Fisher Scientific, Waltham, MA, USA) supplemented with 2 mM L-glutamine, antibiotic/antimycotic (all supplements from Gibco) and 10% heat-inactivated fetal bovine serum (Atlanta Biologicals, Inc. Flowery Branch, GA, USA) and kept in an incubator at 37 °C and 5% $CO_2$. Cells were routinely tested for mycoplasma contamination (MycoAlert® Mycoplasma Detection Kit, Lonza). Cells were seeded on 60 cm$^2$ dishes at $4 \times 10^6$ cells/dish 24 h before transfection. The indicated amount of cDNA was transfected into HEK-293 cells using polyethylenimine (PEI; Sigma-Aldrich) in a 1–2 DNA:PEI ratio. Cell harvesting for radioligand binding experiments or signaling assays were performed approximately 48 h after transfection.

**Radioligand binding assays**. HEK-293 cells were transfected with 5 μg/dish of AAV packaging plasmids encoding for hM3Dq (Addgene #89149), hM4Di (Addgene #89150) or a control vector and harvested 48 h after transfection. Cells were suspended in Tris-HCl 50 mM pH 7.4 supplemented with protease inhibitor cocktail (1:100, Sigma-Aldrich, St. Louis, MO, USA). The dissected brain tissue was diluted in Tris-HCl 50 mM buffer supplemented with protease inhibitor cocktail (1:1000). HEK-293 cells or brain tissue were disrupted with a Polytron homogenizer (Kinematica, Basel, Switzerland). Homogenates were centrifuged at 48,000$g$ (50 min, 4 °C) and washed twice in the same conditions to isolate the membrane fraction. Protein was quantified by the bicinchoninic acid method (Pierce). For competition experiments, membrane suspensions (50 μg of protein/ml) were incubated in 50 mM Tris-HCl (pH 7.4) containing 10 mM $MgCl_2$, 2.5 nM of [$^3$H] clozapine (83 Ci mmol$^{-1}$, Novandi Chemistry AB, Södertälje, Sweden) and increasing concentrations of the competing drugs during 2 h at RT. Nonspecific binding was determined in the presence of 10 μM clozapine. In all cases, free and membrane-bound radioligand were separated by rapid filtration of 500-μl aliquots in a 96-well plate harvester (Brandel, Gaithersburg, MD, USA) and washed with 2 ml of ice-cold Tris-HCl buffer. Microscint-20 scintillation liquid (65 μl/well, PerkinElmer) was added to the filter plates, plates were incubated overnight at RT and radioactivity counts were determined in a MicroBeta2 plate counter (PerkinElmer, Boston, MA, USA) with an efficiency of 41%. One-site competition curves were fitted using Prism 7 (GraphPad Software, La Jolla, CA, USA). $K_i$ values were calculated using the Cheng–Prusoff equation.

**In vitro functional assays**. BRET assays were performed to detect receptor ligand-induced $G_{\alpha}o1$ protein activation. HEK-293 cells were transfected with 5 μg/dish of pAAV plasmids encoding for hM3Dq (Addgene #89149), hM4Di (Addgene #89150) or a control vector together with 0.5 μg Gα-Rluc8, 4.5 μg β1 and 5 μg γ2-mVenus/dish. Forty-eight hour after transfection cells were harvested, washed and resuspended in phosphate-buffered saline (PBS). Approximately, 200,000 cells/well were distributed in 96-well plates, and 5 μM Coelenterazine H (substrate for

luciferase) was added to each well. Five minutes after addition of Coelenterazine H, ligands were added to each well. The fluorescence of the acceptor was quantified (excitation at 500 nm and emission at 540 nm for 1-s recordings) in a PheraStar FSX plate reader (BMG Labtech) to confirm the constant expression levels across experiments. In parallel, the BRET signal from the same batch of cells was determined as the ratio of the light emitted by mVenus (510–540 nm) over that emitted by RLuc (485 nm). Results were calculated for the BRET change (BRET ratio for the corresponding drug minus BRET ratio in the absence of the drug) 5 min after the addition of the ligands.

Intracellular $Ca^{2+}$ concentration was monitored using the fluorescent $Ca^{2+}$ biosensor GCaMP6f. HEK-293 cells were transfected with 7 μg/dish of the cDNA encoding for hM3Dq (Addgene #89149) or hM4Di (Addgene #89150) and 7 μg/dish of GCaMP6. Forty-eight hours after transfection, cells were harvested, washed, resuspended in $Mg^{2+}$-free Locke's buffer pH 7.4 (154 mM NaCl, 5.6 mM KCl, 3.6 mM NaHCO$_3$, 2.3 mM CaCl$_2$, and 5 mM HEPES) containing 5.6 mM of glucose and approximately 200,000 cells/well were distributed in black 96-well plates. Increasing concentrations of the indicated compound were added to the cells and fluorescence intensity (excitation at 480 nM, emission at 530 nM) was measured at 18-s intervals during 250 s using a PHERAstar FSX (BMG Labtech). The net change in intracellular $Ca^{2+}$ concentration was expressed as $F - F_0$ where $F$ is the fluorescence at a given concentration of ligand and $F_0$ is the average of the baseline values (fluorescence values of buffer-treated wells).

**Autoradiography**. Flash frozen tissue (both rodents and the monkey) was sectioned (20 μm) on a cryostat (Leica, Germany) and thaw mounted onto ethanol-washed glass slides. Slides were pre-incubated (10 min, RT) in incubation buffer (50 mM Tris-HCl pH 7.4 with 10 mM of $MgCl_2$), then slides were incubated (60 min) in incubation buffer containing [$^3$H]clozapine (3.5 nM), [$^3$H]C21 (10 nM, 41 Ci/mmol, Novandi, Sweden) or [$^3$H]C13 (3.5 nM, 13 Ci mmol$^{-1}$, Novandi, Sweden) with or without increasing amounts of the indicated cold ligands (Tocris (clozapine, C21) or custom synthesis). Slides were air dried and placed in a Hypercassette™ (Amersham Biosciences) and covered with a BAS-TR2025 Storage Phosphor Screen (FujiFilm, Japan). The slides were exposed to the screen for 5–7 days and imaged using a phosphor imager (Typhoon FLA 7000; GE Healthcare).

**[$^{35}$S]GTPγS autoradiography**. Flash frozen tissue was sectioned (10 μm) on a cryostat (Leica) and thaw mounted on ethanol cleaned glass slides. Sections were encircled with a hydrophobic membrane using a PAP pen (Sigma-Aldrich). Pre-incubation buffer was pipetted onto each slide and allowed to incubate for 20 min (50 mM Tris-HCl, 1 mM EDTA, 5 mM $MgCl_2$ and 100 mM NaCl). The pre-incubation buffer was removed via aspiration and each slide was loaded with GDP in the presence of DPCPX and allowed to incubate 60 min (Preincubation buffer, 2 mM GDP, 1 μM DPCPX, Millipore water). GDP buffer was removed via aspiration and [$^{35}$S]GTPγS cocktail (GDP buffer, 1.3 mM DTT, 2.7 mM GDP, 1.3 μM DPCPX, 83 pM [$^{35}$S]GTPγS) with agonists of interest (C21 10 nM, clozapine 10 nM), without agonists (basal condition), or with a saturated concentration of nonradioactive GTP (for nonspecific binding) was pipetted onto each slide and allowed to incubate for 90 min. The [$^{35}$S]GTPγS cocktail was removed via aspiration and slides were washed in ice-cold washing buffer (50 mM Tris-HCl, 5 mM $MgCl_2$, pH 7.4) for 5 min (2×) followed by a 30 s dip in ice-cold deionized water. Hydrophobic membrane was removed with a cotton swab and xylene and slides were placed into a Hypercassette™ covered by a BAS-SR2040 phosphor screen (FujiFilm; GE Healthcare). The slides were exposed to the phosphor screen for 3–5 days and imaged using a phosphor imager (Typhoon FLA 7000; GE Healthcare).

**Binding and enzyme target profile screen**. These experiments were performed by an outside vendor (Eurofins, France). Briefly, membrane homogenates from stable cell lines expressing each receptor/enzyme were incubated with the respective radioligand in the absence or presence of clozapine or C21 or reference control compounds in a buffer. In each experiment, the respective reference compound was tested concurrently with the test compound to assess the assay reliability. Nonspecific binding was determined in the presence of a specific agonist or antagonist at the target. Following incubation, the samples were filtered rapidly under vacuum through glass fiber filters presoaked in a buffer and rinsed several times with an ice-cold buffer using a 48-sample or 96-sample cell harvester. The filters were counted for radioactivity in a scintillation counter using a scintillation cocktail.

**P-glycoprotein (P-gp) substrate assay**. These experiments were performed by an outside vendor (Eurofins, France). C21 was tested in P-gp substrate assessment assays at 10 μM. The A to B and B to A permeability was measured in Caco-2 cells in the presence and absence of verapamil, a P-gp inhibitor. Efflux ratios ($E$) were calculated based on the apparent B–A and A–B permeability with and without verapamil. In each experiment, the respective reference compound was tested concurrently with the test compound to assess the assay reliability. Fluorescein was used as the cell monolayer integrity marker. Fluorescein permeability assessment (in the A–B direction at pH 7.4 on both sides) was performed after the permeability

assay for the test compound. The cell monolayer that had a fluorescein permeability of less than $1.5 \times 10^{-6}$ cm s$^{-1}$ for Caco-2 was considered intact, and the permeability result of the test compound from intact cell monolayer was reported.

**Bioanalytical methods**. Monkey blood and CSF samples were collected from totally implanted subcutaneous access ports (Access Technologies, Virginia), connected to catheters indwelling in the femoral artery or intrathecal space of the spinal column, respectively. Rodent blood samples and brains were collected immediately following sacrifice at the indicated time points after intraperitoneal injection (10 ml kg$^{-1}$) in buffered saline. CSF was immediately frozen on dry ice and stored at $-80$ °C. Blood samples were allowed to coagulate for 15 min and then centrifuged at 4 °C for 15 min. Serum was collected from the supernatant and stored at a minimum of $-30$ °C until extraction. To 25 µl of serum, 5 µl of internal standard and 110 µl of methanol were added. Samples were centrifuged for 10 min at 16,200×$g$ at 4 °C and the supernatant was transferred to the autosampler vial for analysis. Brains were cut in half and weighed prior to sample preparation. Half brains were homogenized in 490 µl of 85% ethanol: 15% water containing 0.1% formic acid and 5 µl of internal standard using a polytron homogenizer and centrifuged for 10 min at 16,200×$g$ at 4 °C. A 300 µl of supernatant was dried under a stream of nitrogen and resuspended in 150 µl methanol. The resuspended solution was then centrifuged and 100 µl of supernatant was transferred to the autosampler vial for analysis.

Data were acquired using a Nexera XR HPLC (Shimadzu) coupled with a QTRAP 6500 (SCIEX), and was analyzed with Analyst 1.6 (SCIEX). The positive-ion mode data were obtained using multiple reaction monitoring (MRM). The instrumental source setting for curtain gas, ion spray voltage, temperature, ion source gas 1, and ion source gas 2 were 30 psi, 5500 V, 500 C, 650 psi, and 5560 psi, respectively. The collision activated dissociation was set to medium and the entrance potential was 10 V. C21 was monitored using the MRM ion transition $(278.80 \rightarrow 90.0)$ with declustering potentials (DP) = 90 V, collision cell exit potentials (CXP) = 10 V and collision energies (CE) = 50 V. JHU37160 and JHU37152 were monitored using the MRM ion transitions $(359.10 \rightarrow 288.10)$ with DP = 70 V, CXP = 8 V and CE = 28 V. Clozapine was monitored using the MRM ion transitions $(327.30 \rightarrow 270.10)$ with DP = 100 V; 80 V, CXP = 11 V and CE = 40 V.

Separation of the C21, JHU37160, JHU37152, and clozapine was accomplished using a C18 Security guard cartridge ($4.6 \times 4$ mm) and an Eclipse XDB-C18 column ($4.6 \times 250$ mm, 5 µm, Agilent) at 35 °C. Mobile phase A consisted of water containing 0.1% formic acid and mobile phase B was methanol containing 0.1% formic acid. The following linear gradient was run for 21.0 min at a flow rate of 0.4 ml min$^{-1}$: 0–2.00 min 20% B, 7.0 min 80% B, 12 min 90% B, 18.0 min 90% B, 18.1 min 20% B. Twelve-point calibration curves were prepared in standard solution by a 0.5 serial dilution of standards from 0.92 µg ml$^{-1}$ for C21; 1 µg ml$^{-1}$ for JHU37160; and 0.2 µg/ml for JHU37152 and 0.4 µg ml$^{-1}$ for clozapine. The injection volume per sample was 10 µl. Samples were kept at 4 °C in the autosampler tray prior to injection.

The data were measured using standard curves and quality controls, but it was not validated to ICH guidelines. The concentrations of C21, JHU37160, and JHU37152 was measured using area ratios calculated with the internal standard clozapine (5 µl of 100 µg ml$^{-1}$) and the concentrations of clozapine was measured using area ratios calculated with JHU37160 as the internal standard (5 µl of 50 µg ml$^{-1}$). Quality control standards (low, middle and high) were prepared by adding the spiking standard to solution to 25 µl of serum and/or a half-brain and relative values are reported.

**Adeno-associated virus (AAV) injections in rodents**. Animals were anesthetized with isoflurane or a mix of ketamine/xylazine and prepped on a stereotaxic apparatus (Kopf, Germany). The following AAVs expressing hM4Di or hM3Dq fused to mCherry (or EGFP as a control) under the control of a hSyn promoter were used when indicated: hSyn-hM4D(Gi)-mCherry (Addgene: 50475-AAV8), hSyn-hM3D(Gq)-mCherry (Addgene: 50474-AAV8), hSyn-DIO-hM3D(Gq)-mCherry (Addgene: 44361-AAV8), and hSyn-EGFP (Addgene: 50465-AAV8). Based on corresponding mouse and rat brain atlases (Paxinos and Watson), the following coordinates were used to target: the dorsal striatum: Mouse—AP = 1.00, ML = ±1.50, DV = $-3.55$; Rat—AP = 1.70, ML = ±2.50, DV = $-5.50$, rat cortex: AP = 1.70, ML = ±2.50, DV = $-3.50$, Rat VTA: AP = $-5.5$, ML = ± 0.8, DV = $-8.2$. In all cases 1 µl/side was injected and all injections were performed using a Hamilton Neuros 33G syringes at a flow rate of 50 nl min$^{-1}$, except for the injections targeting the rat VTA, that were injected with a picospritzer over 90 s.

**Lentivirus injections in monkeys**. Surgical procedures were performed in a veterinary operating facility under aseptic conditions. Vital signs were monitored throughout the procedure. A pre-operative T1-weighted magnetic resonance imaging (MRI) for each monkey was used to determine the stereotaxic coordinates for the sites of the lentivirus injection in the right amygdala. The skull region above the target site was exposed by retracting the skin, fascia, and muscle in anatomical layers. A small region of cranial tissue was then removed (~1.5 cm diameter) to access the dura mater, into which incisions were made to provide access for the infusion apparatus.

Lentivirus expressing an hM4Di-CFP fusion protein under an hSyn promoter[11] with a titer of >10$^9$ infectious particles was loaded into a 100 µL glass syringe (Hamilton Co., MA). The 31-gauge needle of the syringe was sheathed with a silica capillary (450 µm OD) to create a step 1 mm from the base of the aperture. The syringe was mounted in a Nanomite pump (Harvard Apparatus, Cambridge, MA). The needle was lowered through the incision in the dura mater to each of the pre-calculated target sites and 10 or 20 µL was infused at a rate of 1 µL min$^{-1}$. Each monkey received a total of between 10 and 12 injections (for a total injection volume of 120–240 µL). Post-infusion, the needle was left in situ for 10 min after each injection to allow pressure from the infusate to dissipate. The needle was then slowly removed. At the completion of the injection series the soft tissues were sutured together in anatomical layers.

**Immunohistochemistry**. Rodents were anesthetized with a ketamine and xylazine mixture and transcardially perfused with PBS followed by 4% paraformaldehyde (PFA). The brains were post-fixed in 4% PFA (overnight, 4 °C) and then placed in 30% sucrose for 3–4 days. The brains were frozen and sectioned on a cryostat (40 µm) and collected in PBS with 0.1% Tween-20 (washing buffer). Slices were blocked with bovine serum albumin 3% in washing buffer (blocking buffer, 2 h RT), and incubated the primary antibody mixture: chicken anti-GFP (1:400, ab13970, Abcam Inc.) and rabbit anti-HA (1:400, C29F4, Cell Signaling Technologies) overnight at 4 °C. Sections were washed in washing buffer ($3 \times 10$ min, RT) and incubated with the secondary antibody mix: Alexa488-conjugated goat anti-rabbit (1:200, A11034, Invitrogen) and Alexa546-conjugated goat anti-chicken (1:400, A11040, Invitrogen) and To-Pro3 iodide (Invitrogen) as a nuclear counterstain. After washing ($3 \times 10$ min in washing buffer and $1 \times 5$ min in PBS) slices were mounted on glass slides. Alternatively, flash frozen tissue was sectioned (20 µm) on a cryostat (Leica) and mounted on ethanol-soaked glass slides. Sections were fixed with PFA (4%, 10 min at RT), permeabilized with PBS with TritonX-100 (0.1%, washing buffer), blocked with bovine serum albumin 5% in washing buffer (2 h, RT), and then incubated overnight at 4 °C with the primary antibodies mixture: rabbit anti-mCherry (1:500, ab167453, Abcam Inc.) and chicken anti-GFP (1:2000, ab13970, Abcam Inc.). Sections were then washed again and incubated with Alexa 647 goat anti-rabbit (1:200, A21245, Invitrogen) and Alexa 488 goat anti-chicken (1:400, A11039, Invitrogen) and washed again. In both cases, sections were coverslipped using mounting medium (ProLong Diamond antifade mountant, Invitrogen), and images were acquired either using a confocal microscope (Examiner Z1, Zeiss, Germany) with a laser scanning module (LSM-710, Zeiss, Germany) or a Leica Zoom.V16 stereo microscope (Leica, Germany).

**Synthesis of [11C]clozapine and [18F]JHU37107**. [11C]clozapine was synthesized using the methods developed by Bender et al.[17] with minor modifications. Briefly, 1 mg of *N*-desmethylclozapine was dissolved in 200 µL of acetonitrile. [11C]Methyl triflate was bubbled into the solution until the radioactivity reached a plateau. The reaction was kept at room temperature for 2 min. The solution was then diluted with 200 µL of 40:60 (v:v) acetonitrile:water (0.1% ammonium hydroxide) and injected onto semi-preparative HPLC. The column (Waters XBridge C$_{18}$ 10 mm × 150 mm) was eluted with 40:60 (v:v) acetonitrile:water (0.1% ammonium hydroxide) at a flow rate of 10 mL min$^{-1}$. The radioactive peak corresponding to [11C] clozapine ($t_R = 6.1$ min) was collected in a reservoir containing 50 mL of water and 250 mg of L-ascorbic acid. The diluted product was loaded onto a solid phase extraction cartridge (Waters Oasis HLB plus light) and rinsed with 3.0 mL of water. The product was eluted with 400 µL of ethanol into a sterile, pyrogen-free bottle and diluted with 4.0 mL of saline. A 10 µL aliquot of the final product was injected onto an analytical high-performance liquid chromatography (HPLC) column (Waters XBridge C$_{18}$ 4.6 mm × 100 mm) and eluted with 35:65 (v:v) acetonitrile: water (0.1% ammonium hydroxide) at a flow rate of 2 mL min$^{-1}$. The radioactive peak corresponding to [11C]clozapine ($t_R = 9.2$ min) coeluted with a standard sample. The semi-preparative HPLC eluant used was 25:75 (v:v) acetonitrile:water (0.1% ammonium hydroxide), and no ascorbic acid was added to the reservoir. The molar activity for both [11C]clozapine ranged from 351 to 483 GBq µmol$^{-1}$ (9482–13,041 mCi µmol$^{-1}$) at end of synthesis.

[18F]JHU37107 was prepared via the no-carrier-added $^{18}$F-fluorination using an FDG Nuclear Interface module (Muenster, Germany). Briefly, 4 mg precursor (8-chloro-11-(4-ethylpiperazin-1-yl)-5H-dibenzo[b,e][1,4]diazepin-2-ol) and 8 mg tris(acetonitrile)cyclopentadienylruthenium(II)hexafluorophosphate (STREM, Boston, MA) were dissolved in ethanol (0.3 mL), the solution was heated at 85 °C for 25 min and evaporated to dryness under a stream of argon gas. The residue was dissolved in DMSO (0.5 mL) and acetonitrile (0.5 mL) and the solution was added to a dry complex of [18F]fluoride and 9 mg 1,3-bis(2,6-di-i-propylphenyl)-2-chloroimidazolium chloride (STREM). The reaction mixture was heated at 130 °C for 30 min, diluted with mixture of 0.5 mL acetonitrile, 0.5 mL water and 0.03 mL TFA and injected onto a semi-preparative HPLC column (Luna C18, 10 micron, 10 mm × 250 mm) and eluted with 23:77 (v:v) acetonitrile:water (0.1% trifluoroacetic acid) at a flow rate of 10 mL min$^{-1}$. The radioactive peak corresponding to [18F]JHU37107 ($t_R = 9.9$ min) was collected in a reservoir containing 50 mL of water and 3 mL aq 8.4% NaHCO$_3$ solution. The diluted product was loaded onto a solid phase extraction cartridge (Waters Oasis HLB plus light) and rinsed with 10 mL sterile saline. The product was eluted with 1000 µL of ethanol through a sterile 0.2 µm filter into a sterile, pyrogen-free vial and 10 mL

saline was added through the same filter. The final product [18F]JHU37107 was then analyzed by analytical HPLC (Luna C18, 10 micron, 4.6 mm × 250 mm; mobile phase 23:77 (v:v) acetonitrile:water (0.1% trifluoroacetic acid), flow rate of 3 mL min$^{-1}$; $t_R$ = 6.4 min) using a UV detector at 254 nm to determine the radiochemical purity (>95%) and specific radioactivity (152–188 GBq μmol$^{-1}$ (4100–5070 mCi μmol$^{-1}$)) at the time synthesis ended.

**[11C]clozapine and [18F]JHU37107 imaging using PET**. Mice and rats were anesthetized with isoflurane and placed in a prone position on the scanner bed of an ARGUS small animal PET/CT (Sedecal, Spain) or a nanoScan PET/CT (Mediso, USA) injected intravenously (~100–200 μL) with [11C]clozapine (~700 μCi) or [18F]JHU37107 (~350 μCi) and dynamic scanning commenced. When indicated, animals were pretreated with vehicle or the indicated drug 10 min before the injection of the PET radiotracer. Total acquisition time was 60 min.

All macaque studies were acquired dynamically on the Focus 220 PET scanner (Siemens Medical Solutions, Knoxville, TN). The Focus 220 is a dedicated preclinical scanner with a transaxial FOV of 19 cm and an axial FOV of 7.5 cm. Image resolution is <2 mm within the central 5 cm FOV.

After initial evaluation the monkey was sedated with ketamine (10 mg kg$^{-1}$) followed by ketoprofen (as an analgesic) and glycopyrrolate (for salvia reduction), all weight dependent IM injections. The monkey would then be placed in the supine position, intubated with a tracheal tube. Anesthesia was maintained by 1–3.5% isoflurane and oxygen, the monkey's head was positioned and immobilized for optimal positioning of the brain and moved into the scanner. A 10-min transmission scan using a Co-57-point source for attenuation correction was performed. One iv was inserted, if possible, in the right arm and one in the right leg for injection of tracer and blocking agent. The monkey was always monitored while anesthetized. HR, BP, O$_2$ saturation, RR, three lead ECG, rectal temperature was documented every 15 min.

Arterial blood sampling was acquired throughout the study using an indwelling femoral port. The first 2 min samples were collected every 15 s then at 3, 5, 10, 30, 60, 90, and 120 min post tracer injection. All scans were acquired for 120 min using list mode acquisition.

Scan data were histogrammed into 33 frames (6 × 30 s, 3 × 1 min, 2 × 2 min, and 22 × 5 min). Reconstruction was performed by Filtered Back Projection with scatter correction. After completion of the last study of the day, isoflurane was cut off. The monkey was gradually awakened, moved to the housing facility and fully recovered.

In all cases, the PET data were reconstructed and corrected for dead-time and radioactive decay. All qualitative and quantitative assessments of PET images were performed using the PMOD software environment (PMOD Technologies, Zurich Switzerland). Binding potential $BP_{ND}$ (a relative measure of specific binding) was calculated using a reference tissue model using the cerebellum as a reference tissue in rodents. In macaques, the kinetic data were fitted to a two-tissue compartment model and the concentration of parent in plasma was used as an input function, then the volume of distribution ratios compared to cerebellum were calculated to establish $BP_{ND}$. In all cases, the dynamic PET images were coregistered to MRI templates and time-activity curves were generated using predefined volumes of interest (macaques) or manually drawn in rodents and the described analyses were performed. Receptor occupancy was calculated using the formula: occupancy = $(BP_{ND}^0 - BP_{ND}^{Drug})/BP_{ND}^0 \times 100$, where $BP_{ND}^0$ is the binding potential of the baseline condition and $BP_{ND}^{Drug}$ the binding potential when the animals were pretreated with the drug. In an independent manner, $BP_{ND}$ parametric maps were generated by pixel-based kinetic modeling using a multilinear reference tissue model[18] using the cerebellum as a reference region and the start time ($t^*$) was set to 16 min.

To determine the arterial input function for the radiotracers injected in rhesus monkeys radioligand concentrations in the arterial plasma were corrected by the unchanged parent fraction. Heparinized blood samples (0.5 mL each) were drawn at 15-s intervals until 2 min, and at 3, 5, 10, and 30 min followed by 3-mL samples at 60, 90, 120 min ([11C]clozapine) and 150, 180 min ([18F]JHU37107). Blood samples were immediately sampled for gamma counting, and plasma harvested by centrifugation for gamma counting and radio-HPLC analysis. The unchanged plasma parent fractions were determined by radio-HPLC on an X-terra® $C_{18}$ column (10 μm, 7.8 mm × 300 mm, Waters Corp., Milford, MA), and eluted with MeOH:H$_2$O:Et$_3$N (80:20:0.1; by volume) at an isocratic flow rate of 4.0 mL min$^{-1}$. Eluates were monitored with an in-line flow-through NaI$_{(Tl)}$ scintillation detector (Bioscan). Data were stored and analyzed on a PC using the software "Bio-ChromeLite". The collection of data for each radiochromatogram was decay corrected according to its respective HPLC injection time. Plasma free fraction was determined according to our previous methods[19].

**DREADD-assisted metabolic mapping (DREAMM)**. Mice (D1-hM3Dq, D1-hM4Di or WT littermates, see above) were habituated to experimenter handling and fasted 16 h before the experiment. On the day of the experiment, mice received an IP injection of vehicle (1 ml kg$^{-1}$) and were placed back into their home cages. Ten minutes later, mice were injected (IP) with 11 MBq of 2-deoxy-2-[18F]fluoro-D-glucose (FDG, Cardinal Health) and placed back into their home cages. After 30 min, mice were anesthetized with 1.5% isoflurane, placed on a custom-made bed of a nanoScan small animal PET/CT scanner (Mediso Medical Imaging Systems) and scanned for 20 min on a static acquisition protocol, followed by a CT scan. One

week later the animals were fasted overnight, the next day received an IP injection of C21 (1 mg kg$^{-1}$), clozapine (0.1 mg kg$^{-1}$), or JHU37160 (0.1 mg kg$^{-1}$), and the FDG-PET procedure was conducted as described above. In all cases, the PET data were reconstructed and coregistered to an MRI template as described above. Voxel-based repeated measures with Student's $t$ test comparing baseline to drug were performed, and the resulting parametric images were filtered for statistically significant ($p < 0.05$) clusters larger than 100 contiguous voxels. All statistical parametric mapping analyses were performed using Matlab R2016 (Mathworks) and SPM12 (University College London).

**[3H]clozapine, [3H]C21 and [3H]C13 biodistribution and uptake**. Mice were injected (IP) with [3H]clozapine, [3H]C21, or [3H]C13 (2 μCi g$^{-1}$), euthanized 30 min later and brain, blood, and tissues were collected for radiometric analyses. The brains were flash frozen in isopentane (Sigma-Aldrich) and stored at −80 °C until use. The blood was centrifuged (13,000 rpm, 10 min at RT) and serum was collected. The tissues were solubilized with Solvable™ (PerkinElmer) and bleached with hydrogen peroxide (Sigma-Aldrich). Serum and tissue samples were dissolved in scintillation cocktail and radioactivity counts were determined in a Beckman LS 60000TA scintillation counter (BeckmanCoulter, Indianapolis, IN, USA). Samples were not further analyzed to verify that the tritium signal corresponds to the parent compound so some radiometabolites might contribute to the signal detected in some organs.

**Locomotor activity assessment in mice**. Transgenic male and female mice (see above) (20–30 g) expressing hM3Dq or hM4Di and mCitrine reporters (or controls) were tested for locomotor activity. Mice were injected (IP) with the indicated dose of clozapine, C21, JHU37152 or JHU37160 or vehicle (buffered saline). Ten minutes after injection, animals were placed in an open field arena (Opto-varimex ATM3, Columbus Instruments) and their locomotor activity was tracked during 60 min as infrared beam crossings and traveled distance was converted to cm. Animals were repeatedly tested on consecutive sessions (after an initial habituation session with no drug treatment) in a counterbalanced design.

**Locomotor activity assessment in rats**. Tyrosine hydroxylase (TH):Cre rats ($n = 9$; founder, K. Deisseroth lab) or WT littermates ($n = 9$) were bilaterally injected with AAV2 hSyn-DIO-hM3Dq-mCherry (1ul/hemisphere, Addgene), and allowed to recover for 2 + weeks. Following handling, they were habituated to a Med Associates locomotor testing box (43 × 43 × 30.5 cm) for 2 h on 2 days, then repeatedly tested after separate day doses of DREADD agonists and vehicle. Cohort one was administered C21 (1 or 5 mg kg$^{-1}$) or clozapine (0.001–0.5 mg kg$^{-1}$), and vehicle, in counterbalanced order on separate days, with each test separated by at least 48 h. The 2 h locomotor testing session commenced 30 min after each IP injection on each day. Vehicle was administered on 2 separate days, and average locomotion was used for comparison with DREADD agonists. Cohort 2 underwent the same procedures, but with a 4 h testing session, and the following IP injections: vehicle (x2), JHU37152 (0.01–0.3 mg kg$^{-1}$), JHU37160 (0.01–0.3 mg kg$^{-1}$). All drugs were dissolved in 5% DMSO in saline, and injected 1 ml kg$^{-1}$. Following all tests, rats were transcardially perfused, brains sectioned coronally at 40 μm, and VTA-specific expression confirmed using endogenous mCherry reporter expression, which co-localized nearly exclusively with TH immunoreactivity.

**ChrimsonR generation and characterization**. pcDNA3.1(+)/Hygro vector (Life Technologies, Carlsbad, CA, USA) was used for expression of Chrimson, oC-Chrimson and oC-Chrimson-ts in HEK293 cells (Life Technologies, Carlsbad, CA, USA). Channelrhodopsin inserts (with and without modification) were cloned into the BamHI and XhoI site and the fluorescent protein (tdTomato or citrine) was cloned in-frame and a 3′ stop codon between the XhoI and XbaI site. For neuron expression, the channelrhodopsin-FP inserts were placed in an AAV2 vector between BamHI and HindIII sites (Addgene #50954). Transfection in HEK293 cells were achieved with Fugene (Roche, Basel, Switzerland) and electroporation (Lonza, Gaithersburg, MD, USA) was used for expression in neuronal culture. HEK293 cell experiments were performed 2 days post transfection, whereas neuronal experiments were performed >10 days post transfection. Cell culture conditions were as described previously[20].

For visualizing the channelrhodopsin-FP expression, images were taken on a Zeiss Axiovert 200 M microscope (Zeiss, Jena, Germany) with Slidebook software (3i, Denver, CO, USA) using a Cascade II 1024 EMCCD camera (Photometrics, Tucson, AZ, USA). Images were taken with a 40× oil objective with NA of 1.2. Citrine images were acquired with 495/10 excitation filter and 535/25 nm filters and tdTomato images were acquired with 580/20 nm excitation and 653/95 nm emission filters (Semrock, Rochester, NY, USA). For analyzing membrane/cytosolic fluorescence, a line profile was drawn across the cell in ImageJ or Fiji, the fluorescence intensity of the two membrane regions and cytosolic portion (including nucleus) were measured from profile. After background subtraction, the mean membrane and mean cytosolic fluorescence were calculated and the ratio was calculated.

Electrophysiological recording was performed with an extracellular solution containing 118 mM NaCl, 3 mM KCl, 2 mM CaCl$_2$, 1 mM MgCl$_2$, 10 mM 4-(2-hydroxyethyl)-1-piperazineethanesulfonic acid (HEPES), 20 mM glucose (pH

7.35, 310 mOsm), and an intracellular solution containing 110 mM Cs-methanesulfonate, 30 mM tetraethylammonium chloride, 10 mM ethylene glycol tetraacetic acid (EGTA), 10 mM HEPES, 1 mM CaCl$_2$, 1 mM MgCl$_2$, 2 mM Mg-ATP and 0.15 mM Na$_3$-GTP (pH 7.25, 285 mOsm). All chemicals were acquired from Sigma-Aldrich (St. Louis, MO). Due to the desensitization of the Chrimson response to high intensity of light to some wavelengths, a 1 s 410 nm conditioning light was used to illuminate the recorded cell 10 s prior to testing with indicated wavelength of light.

**In vivo electrophysiology**. Mice were purchased from Janvier Labs and stored in grouped cages (maximun 4 per cage) with ad libitum access to food and water, and in a 12 h light/dark cycle (lights on at 7:00 a.m.). All the experiments were performed in accordance to the national Danish law for the use of laboratory animals and approved by local authorities.

For AAV injection, mice were anaesthetized with Isoflurane and placed in a stereotaxic holder (Narishige, Japan). A trephine hole was drilled above the MGM (from bregma, AP 3.1 mm, ML 1.8 mm). Several injections of a mixture of *ssAAV-8/2-hSyn1-hM4D(Gi)_mCherry-WPRE-hGHp(A)* and *ssAAV-8/2-hSyn1-oChIEF_ChrimsonR_mCitrine-WPRE-SV40pA* (in a 1:4 viral particle ratio) were performed sequentially in different locations (AP 3.05 mm, ML 1.75 mm, DV 3.2 and 3.5 mm; second location AP 3.2 mm, ML 1.85 mm, DV 3.3 and 3.6 mm) so that a total of 4 injections of 0.5 μL were made to distribute the virus evenly through the desired location. Injections were made using a pressurized picospritzer and a pulled glass pipette. After viral injection, animals were kept for 4 weeks to ensure maximum virus expression through axon terminals.

For recording experiments, animals were anesthetized with 1.8 mg kg$^{-1}$ urethane (i.p.), injected with 0.1 ml lidocaine (s.c.) in the incision points, and placed in a stereotaxic holder (Narishige, Japan). A trephine was drilled above the LA (from bregma, AP 1.5 mm, ML 3.1 mm) and the 32 channel opto-electrode was lowered whilst stimulating until a maximum response was located (DV 4 ± 0.2 mm). Body temperature was maintained constant at 37 °C through a feedback-regulated heating pad. Before recording, tissue was left for 10–20 mins for accommodation. Input–output test curves were recorded for both intensity and pulse length 30 min before and 1 h after drug injections using 1 ms-long pulses at 0.33 Hz (450 nm, 110 mA light intensity, ~14 mW, see Supplementary Fig. 10 Panel A). For electrical stimulation experiments (see Supplementary Fig. 10, Panels F–H), a stimulation bipolar twisted platinum–iridium microelectrode was placed in the internal capsule (AP −1.7 mm, ML 2.5 mm, DV 4.0 mm) to target the axons innervating the amygdala. Electrical pulses were given at an intensity that evoked 80% of the maximum response as biphasic 100 μs long pulses at 0.33 Hz.

Raw data were filtered (0.1–3000 Hz), amplified (100×), digitized and stored (10 kHz sampling rate) for offline analysis, with a tethered recording system (Multichannel Systems, Reutlingen, Germany). Analysis was performed using custom routines. After completion of the experiment, animals were sacrificed by decapitation and brains were extracted and kept in 4% PFA overnight (RT). Brains were then sliced (50 μm) and inspected for viral infection location, extent, and electrode placement.

**Molecular modeling**. The crystal structure of human muscarinic M4 receptor [Protein Data Bank (PDB) code: 5DSG] was used (after removing the bound ligand, tiotropium) for our modeling study. We made the Y113$^{3.33}$G and A203$^{5.56}$G in the M4 structure to create a DREADD (hM4Di). JHU37160 was prepared using LigPrep (Schrodinger LLC: New York, NY, 2017). The p$K_a$ calculations using Epik from Schrodinger suite (release 2017–4) and Chemicalize predicted that the piperazine nitrogen closer to the ethyl moiety has p$K_a$ values of 7.4–7.5 and we therefore protonated that nitrogen. Docking was performed using induced-fit docking protocol[21] (Schrodinger LLC, New York, NY, 2017) with the OPLS3 force field. Four largest docking clusters with various orientations of the dibenzodiazepine moiety in the ligand binding site were identified using Clustering of Conformers in Maestro. The poses with the lowest docking scores from each of these four clusters were chosen for further MD simulations. Desmond MD systems (D. E. Shaw Research, New York, NY) with OPLS3 force field was used for the MD simulations. hM4Di was placed into explicit 1-palmitoyl-2-oleoyl-sn-glycero-3-phosphocholine lipid bilayer using the orientation of the 5DSG structure from the Orientation of Proteins in Membranes database. Simple point charge water model was used to solvate the system, charges were neutralized, and 0.15 M NaCl was added. The total system size was ~96000 atoms. The NPγT ensemble was used with constant temperature (310 K) maintained with Langevin dynamics, 1 atm constant pressure achieved with the hybrid Nose-Hoover Langevin piston method on an anisotropic flexible periodic cell, and a constant surface tension ($x$–$y$ plane). The system was initially minimized and equilibrated with restraints on the ligand heavy atoms and protein backbone atoms, followed by production runs at 310 K with all atoms unrestrained. Two independent trajectories for each of the four JHU37160 poses were collected with an aggregated simulation length of 10.56 μs. Only one of the four poses both retained its ionic interaction with Asp112$^{3.32}$ and showed convergence between the two trajectories, and thus was chosen for our further analysis shown in Fig. 3m.

**Statistics**. Sample sizes were chosen based on our results from previous experiments. Depending on experiment, we used paired/two-sample $t$ tests or single factor and multifactor ANOVAs with Dunnett's or Tukey post hoc tests, taking repeated measures into account where appropriate. All statistical tests were evaluated at the $P \leq 0.05$ level.

**Reporting summary**. Further information on research design is available in the Nature Research Reporting Summary linked to this article.

## Data availability

The source data underlying the figures are provided as a Source Data file or can be obtained from the authors upon request.

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

## Acknowledgements

This work was supported by the NIDA (ZIA000069), NIMH (ZIAMH002793, ZIAMH002795), NIA, and NINDS Intramural Research Programs, the NIA (R21AG054802) to A.G.H., the NIBIB (P41EB024495) to M.G.P. and by a European Research Council (ERC) 679714 STC grant to S.N. and Lundbeck Foundation fellowship R264-2017-3189 to A.M. We thank Takafumi Minamimoto and Yuji Nagai for consultation on how to establish PET imaging of DREADD expression in rhesus monkeys at the NIH, and for validation of initial results. We thank NIMH's Molecular Imaging Branch (Chief, Robert Innis) for imaging the monkeys: Robert Gladding and Jeih-San Liow for assistance with camera operation and data acquisition, Sanjay Telu and Victor Pike for radiochemical synthesis and Sami Zoghbi and Michael Frankland for radio-metabolite analysis. We thank Marisela Morales, Amy Newman, and Sergi Ferré for access to instrumentation, Hirsch Davis for access to pharmacological screening resources and Jian Jin for providing C13, C21, and C22. We thank Robert Dannals, Polina Sysa-Shah, James Engles, Nancy Ator, Taek-Soo Lee, Ben Tsui, and Peter Koncz for access to resources and technical support. We also thank Alex Cummins for assistance with macaque tissue preparation, Walter Lerchner and Violette der Minassian for lentivirus production, J. Megan Fredericks, Janita Turchi and Jalene Shim for compound formulation and administration, and Michael Frankland for assistance with blood sampling.

## Author contributions

All coauthors reviewed the paper and provided comments. J.B., M.A.E., F.H., J.L.G., M.S.S., A.M.A, S.L, M.B., C.R., M.F., S.S.S., S.T., S.S.Z., R.L.G., A.M., I.M.G.F., N.A. and J.Y.L. performed the experiments, chemical synthesis, and/or analyzed data. J.B., M.A.E., F.H., J.L.G., M.S.S., A.M.A, A.M., J.Y.L, V.W.P, R.B.I., R.M., P.M., L.S., D.R.S., S.V.M., S.N., A.G.H., B.J.R. and M.M. designed and/or supervised experiments and syntheses. M.G.P. and A.B. provided access to resources and support. J.B. and M.M. wrote the paper with input from all authors. M.M. conceived the study.

## Additional information

**Competing interests:** M.M. is a cofounder and owns stock in Metis Laboratories. J.B., J.L.G., F.H., M.S.S., A.G.H., M.G.P., and M.M. are listed as inventors on an application (62/627,527) filed with the U.S. Patent Office regarding the novel DREADD compounds described herein. Remaining authors declare no competing interest.

