## [Peer Review File · Nature Communications]

Reviewers' comments:

Reviewer #1 (Remarks to the Author):

The stated purpose of the study was to validate novel DREADD ligands alleged to be superior to CNO or C21 in terms of brain penetrance, affinity, and potency, and which may be more amenable to clinical use. Both in vitro and in vivo rodent and non-human primate experiments were included, and the investigation was very thorough. In my opinion, the highlight of the manuscript is the development of a PET-compatible DREADD radioligand. However, there are several conceptual issues that limit the potential impact of this work.

DREADDs have two applications: as a neuroscience tool to manipulate neuronal activity in animals, and as a potential therapeutic for treating human disease. The authors seem to argue that DREADD technology is in dire need of improvement for both applications. As a research tool, the fact is that CNO, low dose clozapine, and Compound 21 all work quite well. While they certainly have some issues with brain penetrance, pharmacokinetics, and off-target effects, a vast majority of studies have used them successfully for their stated purpose, to manipulate neuronal activity, when appropriate controls are used. Thus, it is not clear how the new DREADD ligands described in this manuscript will be markedly better for neuroscience applications. As far as human therapeutics goes, there are many ongoing efforts to develop better chemogenetic tools, some using completely different types of receptors and ligands (e.g. kappa opioid receptor-based DREADDs with salvinorin B) and some of which employ FDA-approved drugs (e.g. α 7-5HT3 with varenicline). Thus, from a clinical perspective, the main obstacle is not the ligand, but rather getting the chemogenetic receptor into the human brain, which this manuscript does not address. Overall, this is fine work, but does not represent a substantial advance from a basic neuroscience or clinical standpoint. Other weaknesses are noted below.

1. A large fraction of the paper (2 of the 6 main figures) is focused on showing that C21 is not a good DREADD ligand. This feels like overkill and is in some ways misleading. In Fig 2, the authors show that 1 mg/kg C21 is as efficacious as 0.1 mg/kg clozapine, which makes C21 a perfectly acceptable DREADD ligand (CNO is frequently used at 1 mg/kg). C21 is behaviorally effective at this dose, which seems incompatible with it having the poor brain penetrance and inability to bind to DREADDs in mouse brains, as the authors suggest. In Fig 1E and 3P-Q, the authors show only negligible differences in potency for clozapine, C21, and the novel compounds in HEK cells.

2. The novel ligands presented in this manuscript have virtually the same undesirable off-target drug actions as clozapine, which is part of the problem with using clozapine or CNO for DREADD experiments in the first place. The data in Fig S4 indicate that J52 and J60 are potent, nanomolar affinity antagonists of 5-HT_{2A}R, D₁Rs, D₂R, D₄Rs, H₁, and all muscarinic receptors. In this same figure, the authors emphasize C21 binding to opioid receptors, which is overstated and mostly applies to the delta OR. Thus, vehicle controls will still be crucial for experiments with J52 and J60, and the claim that the new ligands are a huge improvement over existing compounds does not feel justified.

3. Figure 6L-M was not mentioned in the text.

4. The clozapine binding assay experiments can be interpreted differently. For example, the inability of C21 to displace clozapine binding could be easily explained by the existence of an alternative binding site for C21. C21 may simply not compete for the same site as clozapine on DREADDs. The GTPyS experiments are also not definitive, as clozapine exerts significant partial agonism at a variety of non-DREADD GPCRs.

5. Development of the PET ligand is the most novel and interesting part of the paper, but it is presented at the end in such a way that makes it feel like a side note.

6. It is not clear what "neurotheranostics" means.

Reviewer #2 (Remarks to the Author):

The work from Bonaventura, Eldridge et al. is an important step in tool development for imaging the expression density of certain DREADDs and for functional modulation of DREADDs with new ligands that have improved brain PK properties. The premise of the work was initially based on a compound (C21) that had been previously reported and the authors detail its deficiencies in brain uptake and competition binding assays with DREADDs. Ultimately, the paper details the synthesis and evaluation of new, highly potent DREADD activators (also based on clozapine) and demonstrate occupancy at DREADDs using PET imaging. The work has been expertly performed and the conclusions are well supported by the data, but the manuscript presentation could use a lot of improvement:

- The story of compound 21 (C21) and its failure in the assays/experiments presented are important contributions to the literature but not Nature Communications worthy. I strongly suggest this is separated from the successful development of the better tool compounds and PET imaging ligand (which are Nature Communications worthy). The C21 failure is a self-contained story and it feels very much like the paper starts over on page 7 line 6. The reason for the suggestion is that the JHU compounds are very good, well characterized and show great potential. The authors are diluting the importance of the work by walking the reader through the failure of one compound when they could recount the development of the better ones up front. The stories do not need to be entangled and the reader does not benefit from the story being told in a timeline progression.

Comments on the C21 story:

- The 3H in vivo experiments and conclusions need to be reconsidered in the context of the signal. If HPLC was not used to verify the chemical species of the 3H in each organ, then all the authors can conclude is information related to C21 and its radiolabeled metabolites. This becomes problematic when taking ratios of organs or comparing them given that the two independent measurements do not in fact necessarily represent the same measurement units. Insufficient details were provide to know, but I suspect the chemical nature of the 3H was not determined for each organ. (Also- Why is brain not shown in Supp Fig 3?)

- In the text, when using "poor brain uptake" or "low brain/serum ratios", please provide the quantitative value that drives the assessment. Many readers may find a 0.4 ratio of brain/serum ok for certain drugs. The use of "low" is therefore in need of a reference point and readers may disagree on the appropriate range. This is also very important since a hypothetical compound with no non-specific binding and a very low target expression (or high expression in a very small region), would in fact prove to have a terrible brain:serum ratio and yet still be a great drug or imaging agent. This comment can be generalized to other text in the results and discussion where the authors could be more precise and quantitative with the written presentation/interpretation of results.

Comments on the JHU story:

- The PET imaging experiments are quite convincing and were rigorously performed with appropriate modeling and outcome measures. Given this, do the authors feel there is room to improve or is DREADD imaging (at least for hM4Di) a fully solved problem? A discussion of limitation and areas for improvement (if any) would be welcome. (e.g. Why is the BPnd in non

expressing regions > 0 ?, Fig 6M)

- Figure 6K should be presented on the full scale 0-1.8, not starting from 0.6.

Overall, great work. I enjoyed reading it and look forward to seeing it in press.

Response to reviewers.

We greatly thank the reviewers' for their remarks, as we sincerely think they helped us improve the manuscript and send a clearer message to the reader. We have made changes in the text (highlighted in yellow) and we will here address the reviewer's concerns point by point.

Reviewer #1 (Remarks to the Author):

The stated purpose of the study was to validate novel DREADD ligands alleged to be superior to CNO or C21 in terms of brain penetrance, affinity, and potency, and which may be more amenable to clinical use. Both in vitro and in vivo rodent and non-human primate experiments were included, and the investigation was very thorough. In my opinion, the highlight of the manuscript is the development of a PET-compatible DREADD radioligand. However, there are several conceptual issues that limit the potential impact of this work.

DREADDs have two applications: as a neuroscience tool to manipulate neuronal activity in animals, and as a potential therapeutic for treating human disease. The authors seem to argue that DREADD technology is in dire need of improvement for both applications. As a research tool, the fact is that CNO, low dose clozapine, and Compound 21 all work quite well. While they certainly have some issues with brain penetrance, pharmacokinetics, and off-target effects, a vast majority of studies have used them successfully for their stated purpose, to manipulate neuronal activity, when appropriate controls are used. Thus, it is not clear how the new DREADD ligands described in this manuscript will be markedly better for neuroscience applications. As far as human therapeutics goes, there are many ongoing efforts to develop better chemogenetic tools, some using completely different types of receptors and ligands (e.g. kappa opioid receptor-based DREADDs with salvinorin B) and some of which employ FDA-approved drugs (e.g. $\alpha 7$ -5HT3 with varenicline). Thus, from a clinical perspective, the main obstacle is not the ligand, but rather getting the chemogenetic receptor into the human brain, which this manuscript does not address. Overall, this is fine work, but does not represent a substantial advance from a basic neuroscience or clinical standpoint. Other weaknesses are noted below.

1. *A large fraction of the paper (2 of the 6 main figures) is focused on showing that C21 is not a good DREADD ligand. This feels like overkill and is in some ways misleading. In Fig 2, the authors show that 1 mg/kg C21 is as efficacious as 0.1 mg/kg clozapine, which makes C21 a perfectly acceptable DREADD ligand (CNO is frequently used at 1 mg/kg). C21 is behaviorally effective at this dose, which seems incompatible with it having the poor brain penetrance and inability to bind to DREADDs in mouse brains, as the authors suggest. In Fig 1E and 3P-Q, the authors show only negligible differences in potency for clozapine, C21, and the novel compounds in HEK cells.*

As both reviewers suggested, we have moved all the C21 data to supplementary material and kept only a brief description of the results in the main text. As described in the original text, C21 is behaviorally effective in rodents (especially for hM3Dq) even at low occupancy rates, which can be explained by the supraphysiological receptor expression achieved in these models. However, the window of selectivity displayed at hM4Di is notably lower (even the minimal behaviorally effective dose of 1mg/kg showed changes in brain metabolism in WT mice), although this will vary based on the

experimental model used in each study and hM4Di expression. More importantly, likely due to both pharmacokinetic and pharmacodynamic factors, the window of selectivity for C21 in monkeys, as our experiments clearly illustrate, is too narrow and hence this ligand is certainly not appropriate for this species.

2. *The novel ligands presented in this manuscript have virtually the same undesirable off-target drug actions as clozapine, which is part of the problem with using clozapine or CNO for DREADD experiments in the first place. The data in Fig S4 indicate that J52 and J60 are potent, nanomolar affinity antagonists of 5-HT_{2A}R, D₁Rs, D₂R, D₄Rs, H₁, and all muscarinic receptors. In this same figure, the authors emphasize C21 binding to opioid receptors, which is overstated and mostly applies to the delta OR. Thus, vehicle controls will still be crucial for experiments with J52 and J60, and the claim that the new ligands are a huge improvement over existing compounds does not feel justified.*

We agree with the reviewer that the selectivity of the new ligands is not yet ideal. To stress this concern, we added a comment in the discussion section and have proposed strategies for overcoming such limitations. In vitro binding “selectivity” assays are not entirely predictive of in vivo efficacy and for this reason we made sure to characterize C21, J52 and J60 using an array of different yet complementary approaches, particularly those that could provide a window into their in vivo performance. A key and very important difference between the new ligands we present here and C21, is that they (especially J60) are more potent than C21 in rodents and also have a much wider selectivity window in monkeys.

3. *Figure 6L-M was not mentioned in the text.*

There was an error on the figure references for this section that has been corrected, and the whole section has been edited to convey a clearer message.

4. *The clozapine binding assay experiments can be interpreted differently. For example, the inability of C21 to displace clozapine binding could be easily explained by the existence of an alternative binding site for C21. C21 may simply not compete for the same site as clozapine on DREADDs. The GTPyS experiments are also not definitive, as clozapine exerts significant partial agonism at a variety of non-DREADD GPCRs.*

Although the existence of an alternative (allosteric) binding site for compound 21 cannot be ruled out, both C21 and clozapine, and their derivatives, have the same chemical scaffold with little difference in their substituents. The molecular docking data and the similarity of the binding profile with that of other GPCRs suggests that all of these compounds bind and compete for the orthosteric site of such receptors. In any case, as we discussed in the text, the in vitro calculated affinities and potencies don't seem to be the best predictor of in vivo performance.

The GTPyS is indeed an assay with high chances of getting off-target signal (mostly from Gi-coupled receptors). Our approach to solve this was to quantify the signal in areas with highly localized expression (due to highly-localized AAV-mediated expression which is readily visualized) using the contralateral side as a control for effects on endogenous targets. We have clarified this in the text.

5. *Development of the PET ligand is the most novel and interesting part of the paper, but it is presented at the end in such a way that makes it feel like a side note.*

We have now restructured the paper to make sure this part is more prominent.

6. *It is not clear what “neurotheranostics” means.*

We have now cited a recent review that provides a very detailed description of the term Neurotheranostics.

Reviewer #2 (Remarks to the Author):

The work from Bonaventura, Eldridge et al. is an important step in tool development for imaging the expression density of certain DREADDs and for functional modulation of DREADDs with new ligands that have improved brain PK properties. The premise of the work was initially based on a compound (C21) that had been previously reported and the authors detail its deficiencies in brain uptake and competition binding assays with DREADDs. Ultimately, the paper details the synthesis and evaluation of new, highly potent DREADD activators (also based on clozapine) and demonstrate occupancy at DREADDs using PET imaging. The work has been expertly performed and the conclusions are well supported by the data, but the manuscript presentation could use a lot of improvement:

- The story of compound 21 (C21) and its failure in the assays/experiments presented are important contributions to the literature but not Nature Communications worthy. I strongly suggest this is separated from the successful development of the better tool compounds and PET imaging ligand (which are Nature Communications worthy). The C21 failure is a self-contained story and it feels very much like the paper starts over on page 7 line 6. The reason for the suggestion is that the JHU compounds are very good, well characterized and show great potential. The authors are diluting the importance of the work by walking the reader through the failure of one compound when they could recount the development of the better ones up front. The stories do not need to be entangled and the reader does not benefit from the story being told in a timeline progression.

As also suggested by the other reviewer, we have moved all the C21 data from the main text to the supplementary information. We believe this will now help to focus the attention of the reader to the newly developed compounds and PET ligand. But at the same time, we believe it is important to share the C21 data with the neuroscience community in direct comparison with other ligands. Previous reports only compared C21 with CNO, and hence they were somewhat overstating its potency. In this new version of the text, we mainly stress the failure of C21 in non-human primates comparing it to clozapine.

Comments on the C21 story:

- The 3H in vivo experiments and conclusions need to be reconsidered in the context of the signal. If HPLC was not used to verify the chemical species of the 3H in each organ, then all the authors can conclude is information related to C21 and its radiolabeled metabolites. This becomes problematic when taking ratios of organs or comparing them given that the two independent measurements do not in fact necessarily represent the same measurement units. Insufficient details were provide to know, but I suspect the chemical nature of the 3H was not determined for each organ. (Also- Why is brain not shown in Supp Fig 3?)

We agree with the reviewer. The chemical entities in the radioactive samples were not measured and hence ratios between organs can't be taken, that's why we showed the corresponding data in Supp Fig 3 (now Supp fig 6) as activity per gram of tissue but not as ratios between blood and organs. The only data showed as a ratio was the bioanalytic experiments where cold (non-radiolabeled) compounds were injected systemically and then analyzed in blood or brain. In this case, the chemical entities were confirmed by HPLC. Notably, in the experiment where we injected a trace dose of ³H-C21 we could only detect radioactive signal in the ventricles, not in the DREADD expressing areas (please, see answer to the question below). We added a sentence in the Methods section to clarify this issue.

- In the text, when using "poor brain uptake" or "low brain/serum ratios", please provide the quantitative value that drives the assessment. Many readers may find a 0.4 ratio of brain/serum ok for certain drugs. The use of "low" is therefore in need of a reference point and readers may disagree on the appropriate range. This is also very important since a hypothetical compound with no non-specific binding and a very low target expression (or high expression in a very small region), would in fact prove to have a terrible brain:serum ratio and yet still be a great drug or imaging agent. This comment can be generalized to other text in the results and discussion where the authors could be more precise and quantitative with the written presentation/interpretation of results.

We agree with the reviewer's point of view, and this is especially true in this scenario, where DREADD expression is spatially restricted to a known anatomical location. Hence, we modified the text to make it more precise specifically describing the pharmacokinetic parameters of C21 in comparison to clozapine. The brain/serum ratio per se does not provide whole-picture information and our results, which were derived using many convergent approaches, illustrate unequivocally that C21 has weaker target engagement than clozapine and our newly described compounds. Although we could detect some non-radioactive C21 in the mouse brain, the brain homogenate includes ventricles, vasculature and other non-CNS areas. The radioanalytical data showed that C21 was not actually in the 'brain' or where the DREADDs were expressed. We believe the two approaches are complementary and support our conclusions.

Comments on the JHU story:

- The PET imaging experiments are quite convincing and were rigorously performed with appropriate modeling and outcome measures. Given this, do the authors feel there is room to improve or is DREADD imaging (at least for hM4Di) a fully solved problem? A discussion of limitation and areas for improvement (if any) would be welcome. (e.g. Why is the BPnd in non expressing regions > 0?, Fig 6M)

We are convinced that F¹⁸-JHU37107 is a valid ligand to image DREADDs. It is true that the ligand might label other endogenous sites and the use of F-18 allows for longer data acquisition periods that could help determine DREADD vs off-target binding through kinetic modeling, blocking studies or pre- vs post-transduction scans. In our hands (in this study's data and other ongoing projects) complex experimental designs were not necessary as the DREADD's high focal expression was enough to identify the DREADD containing areas. Nevertheless, we have now added a few sentences regarding this issue in the Discussion section.

- Figure 6K should be presented on the full scale 0-1.8, not starting from 0.6.

We respectfully propose to keep the data presented in this original manner so as to illustrate (same as it is in now Figure 2 for the C-11 Clozapine scans) that we can effectively isolate the DREADD signal from the background. However, we also believe that the full-scale images should be presented so we have now included these in the Supplemental information.

Overall, great work. I enjoyed reading it and look forward to seeing it in press.

Thank you!

REVIEWERS' COMMENTS:

Reviewer #2 (Remarks to the Author):

Overall, the manuscript has been improved, but many points that reviewer 1 has made are unaddressed. Most of my comments were addressed in the response but not in the text itself (e.g. defining brain uptake quantitatively or removing the imprecise comparative language). The PET imaging work is strong and may be better as the sole focus, but otherwise I think the paper is technically sound.